# Understanding the Female Physical Examination in Patients with Chronic Pelvic and Perineal Pain

**DOI:** 10.3390/jcm11247490

**Published:** 2022-12-17

**Authors:** Augusto Pereira, Lucia Fuentes, Belen Almoguera, Pilar Chaves, Gema Vaquero, Tirso Perez-Medina

**Affiliations:** Department of Gynecologic Surgery, Puerta de Hierro University Hospital, 28222 Madrid, Spain

**Keywords:** chronic pelvic pain, chronic perineal pain, exploration, examination, physical exam, central sensitization

## Abstract

(1) Background: The objective was to compare the exploration of chronic pelvic pain syndrome (CPPS) patients in different locations and establish the role of physical examination in CPPS patients. (2) Methods: We reviewed clinical data from 107 female patients with CPPS unresponsive to conventional therapies at Puerta de Hierro University Hospital Madrid, Spain, from May 2018 to June 2022. Patients were classified into three groups: (a) pelvic pain; (b) anorectal pain; or (c) vulvar/perineal pain. (3) Results: Although the demographics of patients with CPPS were different, their physical examinations were strikingly similar. Our study observed a comorbidity rate of 36% and 79% of central sensitization of pain. Seventy-one percent of patients had vulvar allodynia/hyperalgesia. Pain on examination was identified in any pelvic floor muscle, in any pelvic girdle structure, and neuropathic pain in 98%, 96%, and 89%, respectively. Patients with vulvar and perineal pain were more different from the other groups; these patients were younger and had fewer comorbidities and less central sensitization, less anorectal pain, more pain during intercourse, and greater nulliparity (*p* = 0.022; *p* = 0.040; *p* = 0.048; *p* = 0.000; *p* = 0.006; *p* = 0.005). (4) Conclusions: The findings of this study are related to the understanding of the pathophysiology of CPPS. The physical examination confirms the central sensitization of female patients with CPPS, helps us to determine the therapeutic management of the patient, and can be considered as a prognostic factor of the disease.

## 1. Introduction

Chronic pelvic pain syndrome (CPPS) is a multifactorial pain condition primarily affecting the anatomic pelvis, anterior abdominal wall, infraumbilical, umbilical, and lumbosacral back area, and the buttocks [1,2,3,4,5,6]. Chronic pelvic pain is a pathological condition, and its pain intensiveness increasingly results in functional disability that often requires medical attention [7].

According to the latest research on CPPS [8,9,10,11,12,13], there are two phenotypes in women who experience pelvic pain: (a) those who suffer only from pelvic pain (tier 1), defined as focal and with an unrecognized peripheral pain generator, and (b) those who experience pelvic pain plus other symptoms (tier 2), characterized by increased central sensitization of their pain. Moreover, women with this phenotype (tier 2) experience long-standing pain and severe concomitant psychological dysfunction, which impact their visceral pain problems [14]. 

Although there is an absence of specific recommendations for these phenotypes’ diagnosis, CPPS should be established through anamnesis and physical examination of the patient, yet it still seems that the standard examination for gynecologic patients might be insufficient.

Unfortunately, most studies on CPPS lack phenotype stratification, so evidence on prognostic and treatment outcomes is not individually addressed. Nevertheless, phenotypic distinction is known to be beneficial in counseling women with tier 2 pain, as well as in guiding them toward multimodal and prolonged pain management.

The aim of this study is to compare CPPS exploration in groups of women stratified according to their pain locations and establish the clinical usefulness of physical examination in the diagnosis, advice, and management of the CPPS patient. 

## 2. Materials and Methods

### 2.1. Patients

The medical records of 382 consecutive female patients with CPPS who sought consultation at the Puerta de Hierro Majadahonda University Hospital in Madrid, Spain, from May 2018 to June 2022, were reviewed. Ethical approval for this study was granted by the Puerta de Hierro Majadahonda University Hospital Institutional Review Board.

The main inclusion criteria were female patients with CPPS that had been ongoing for at least three months (a time period that was selected in order to encompass the criteria for vulvodynia patients) and who had not responded to conventional therapy. Patients were classified into three groups according to the pain location as a reason for consultation: (a) pelvic pain (pain localized to the anatomic pelvis, anterior abdominal wall at or below the umbilicus) [1,2,3,4,5,6]; (b) anorectal pain [15]; or (c) vulvar, vestibule, clitoris, and/or perineal pain [16]. When more than one pain location existed, patients were assigned according to their most severe pain or the reason for consultation. The exclusion criteria referred to patients with pain in a location other than those indicated above, male patients, or patients with incomplete data. In the end, 107 female patients who met the above criteria were analyzed. All patients were examined by a single pelvic pain expert physician (A.P).

### 2.2. Data Assessed

The data collected included a complete medical history (gynecological, obstetric, and surgical history), comorbidities, pain in neighboring organs (bladder, rectum, and dyspareunia), duration of pain (estimated by the patient as the number of years between the onset of symptoms and the date of first consultation), pain location, and intensity of the pain as measured via the visual analog scale (VAS), in which zero was defined as no pain and ten as the worst pain ever experienced. 

Comorbidities were defined as the concomitant presence of several clinical hypersensitivity syndromes, such as irritable bowel syndrome, fibromyalgia, interstitial cystitis, migraine, endometriosis, anxiety and depression, neuropathies, temporomandibular joint dysfunction, chronic fatigue syndrome, and/or a history of multiple chemical sensitivity [17].

Clinical criteria of central sensitization for patients with chronic pelvic and perineal pain (Convergences PP Criteria Score) were calculated for each patient and a score ≥5/10 was considered suggestive of central sensitization [18].

### 2.3. Exploratory Procedures or Pain Mapping Method

Data from the examination were collected according to the pain mapping method reported in previously published works [19,20], including the following.
(1)An S2–S4 neurological examination:(a)Cotton swab testing of the S2–S4 dermatome and vestibule. The absence of signs and symptoms during the physical examination confirms the integrity of the C-fibers.(b)An evaluation of the motor response of the nerve using the clitoris, bulbospongiosus, and perineal reflexes. Normal motor activity at S2–S4 is indicated by anal sphincter contraction.(c)Tinel’s sign in the sciatic spine area to evaluate the third segment of the pudendal nerve. Pain is reproduced with transrectal compression of the third segment of the PN (Tinel sign) in the sciatic spine and Alcock’s canal.(d)Tinel’s sign at the clitoris to evaluate the dorsal nerve of the clitoris. The clitoris was compressed to locate painful spots.(2)Exploration of the pelvic girdle: bilateral palpation in order to identify painful spots—retropubic, ischiopubic ramus, ischium, sacrospinous ligament, sacrum, and coccyx area.(3)Exploration of pelvic floor muscles [21]:(a)Levator ani muscle (LAM): assessment of painful palpation of the pubococcygeus muscle.(b)Obturator internus muscle (OIM): contracture of the OIM with flexion and external rotation of the hip in the supine decubitus position and transgluteal examination of OIM segments—pelvic (ischium), medium (midpoint between trochanter and coccyx), and gluteal (hip).(c)Piriformis muscle (PM): simultaneous hip external rotation and abdominal flexion is encouraged to reproduce the pain. PM is palpated transgluteally five centimeters above the OIM middle segment.

Patients were asked to rate their level of pain on a scale from 0 (no pain) to 10 (worst pain ever experienced) during the assessment.

### 2.4. Statistical Analysis

Dichotomous or categorical variables were expressed as absolute values and percentages; continuous variables were expressed as the mean and median. The chi-square test was used to estimate the relationship between variables and pain. The relative risk (RR) was calculated. Differences in distributions of dichotomous variables were analyzed using Fisher’s exact test. Differences in distributions of continuous variables were analyzed using the Kruskal–Wallis test. A value of *p* ≤ 0.05 was considered statistically significant. The statistical analysis was performed using R. (R version 3.6.3 (29 February 2020) Copyright (C) 2020 The R Foundation for Statistical Computing).

## 3. Results

### 3.1. Patients

We identified 382 medical records in our hospital’s electronic database of female patients who sought consultation between May 2018 and June 2022. After excluding 275 cases that had missing data needed for the study, a total of 107 patients met all inclusion criteria and were selected for evaluation.

### 3.2. Data Assessed

The median age of the cohort was 41 years (range: 16–72). The median pain duration was 1.5 years (range: 0.3–26), and the median VAS score was 7/10 (range: 1–10). The most common symptoms in neighboring organs were pain during sexual intercourse (83%) followed by urological pain (48%) and proctalgia (39%). A total of thirty-eight patients (36%) had associated comorbidities: endometriosis (30), anxiety and depression (13), irritable bowel syndrome (10), fibromyalgia syndrome (10), interstitial cystitis (6), migraine (5), neuropathies (4), temporomandibular joint dysfunction (1), chronic fatigue syndrome (1), and a history of multiple chemical sensitivity (1). Detailed demographic data on the patients included in each group and the correlations between them are shown in Table 1.

The Convergence PP Criteria score was calculated in 101 patients. Scores of five or more points, suggestive of sensitization to pelvic pain, were found in 80 patients (79%). The median duration of pain since the onset of symptoms for patients with central sensitization was three years (range: 0.3–26), whereas, for patients without central sensitization, the median duration was 1.5 years (range: 0.3–10). The difference between the two groups was statistically significant (*p* = 0.031).

### 3.3. Exploratory Procedures or Pain Mapping Method

A total of 107 female patients were explored. A sensory deficit in the S2–S4 area was found in 71 out of 107 CPPS patients. A total of 22 patients had a pudendal nerve motor deficit and 76 patients had a positive cotton swab test at the vestibule. Exploration of the third segment of the pudendal nerve revealed pain in 90 patients, and 40 patients had pain in the dorsal nerve of the clitoris. Of the 107 patients, 95, 98, and 56 presented with pain localized in the LAM, OIM, and PM, respectively. The most common pelvic girdle structures affected were the retropubic (86%), followed by the coccyx (67%) and sacrospinous ligament (65%).

There were 105 patients (98%) with pain in any pelvic muscle (LAM, OIM, or PM), 103 patients (96%) with pain in any pelvic girdle structure, and 95 patients (89%) with neuropathic pain. The clinical and examination findings of each group and the correlations between them are shown in Table 2.

### 3.4. Statistical Analysis

Demographic differences were found between the patient groups, possibly related to their specific characteristics. The group with vulvar and/or perineal pain had the demographic characteristics that were the most different in our population; these patients were younger and had fewer comorbidities, less central sensitization, less anorectal pain, more pain during intercourse, and greater nulliparity (*p* = 0.022; *p* = 0.040; *p* = 0.048; *p* = 0.000; *p* = 0.006; *p* = 0.005). The results are shown in Table 1. In contrast, the findings obtained during the examination of all patients were much more homogeneous, with no significant differences observed, except for a lower pain ratio in the trigger point in all pelvic muscles in the anorectal pain group (LAM, OIM, and PM). The results are shown in Table 2.

## 4. Discussion

This study demonstrates that although the case history of patients with CPP varies widely, they have physical examinations that are surprisingly very similar. Central sensitization in CPPS patients is quite frequent and usually accompanied by findings of nociceptive pain (somatic and visceral) and neuropathic pain. According to the opinion of a formal consensus of experts [18], the diagnosis of central sensitization in patients with CPPS requires data from the patient’s case history and data collected from a physical examination (at least two of ten criteria: the existence of pelvic trigger points and the diagnosis of allodynia and perineal and/or vulvar hyperalgesia).

The examination of patients with central sensitization seems to confirm the patient’s phenotype (tier 2), reveals the affected compartments, and allows for the proposal of multidisciplinary procedures for pain management [20]. Pain central sensitization guides treatments towards combined therapies for myofascial pain syndrome, as well as neuropathic and visceral pain treatments. Our results, in endometriotic patients unresponsive to conventional therapies, revealed partial improvements in 70% of women [20].

Furthermore, the diagnosis of central sensitization predicts a poor, transient response to conventional treatment and can be considered a predominant risk factor for postoperative pain [18,22,23]. In pain patients who exhibited central sensitization symptoms, Aasvang et al. identified the significant malfunction of both large and small fibers [22]. The conclusion of a systematic review supported the inclusion, prior to surgery, of the assessment of central pain mechanisms such as temporal summation of pressure pain, conditioned pain modulation, and reactions to pain above the pain threshold, as variables associated with pain after surgery [23]. All of this information should be included in patient counseling prior to treatment decision making.

In routine clinical practice, many patients seek a diagnosis for long-standing persistent pelvic pain. However, it seems to be that after taking a detailed case history and performing a physical examination, many possibilities still remain. With many years of knowledge on CPPS, the answer may lie in understanding the pathophysiology of CPPS, particularly in regard to the following.
(a)Pain perception occurs through afferent pain pathways. When tissue damage occurs, the release of local inflammatory mediators occurs, which are detected by nociceptors in response to a noxious inflammatory stimulus. C-fibers can become sensitized if C-fiber nociceptors are activated and, thus, they no longer remain silent, even after resolution of the inflammation [24]. In normal circumstances, peripheral sensitization leads to a decrease in the threshold for neuronal activation, thus resulting in pain when faced with a normally non-painful stimulus (allodynia) or increased sensitivity to painful stimuli (primary hyperalgesia).(b)Central sensitization [25,26] begins with peripheral sensitization and is sustained by continuous noxious stimuli to the central nervous system (CNS). Chronic hyperexcitability of peripheral afferents cause changes in the CNS properties of neurons, unlinking pain from the intensity, duration, or presence of noxious peripheral stimuli, as in acute nociceptive pain. At this point, the pain persists long after resolution of the cause, as a pain “memory effect”. Once central sensitization is established, alterations occur in sensory processing in the brain, pain suppression mechanisms malfunction, neurons experience increased excitability, which may exacerbate the perception of pain (secondary hyperalgesia), and a long-term enlargement occurs in neuronal synapses in the cerebral cortex.

The case history for a patient with central sensitization can be complex and misleading. Indeed, the perception of pain can be misinterpreted by both the patient and the physician. Symptoms include burning, tingling, prickling, hyperalgesia, and perineal allodynia. Both the physical examination and case history of these patients can also be unclear. Multiple tender points may be identified, such as a pain response to pelvic nerve stimulation; muscle trigger points; or pain from visceral, perineal, and/or vulvar compression.
(c)The splanchnic, pelvic, and pudendal nerve pathways innervate the female reproductive tract. This is an opportunity for cross-talk through shared neuronal pathways with other neighboring organs. Cross-organ sensitization occurs when sensitized afferents from one organ induce the sensitization of another. Patients may have symptoms [27,28] ranging from pain to dysfunction, and a number of pelvic organs may be affected, such as pain on bladder filling, pain during or after urination, discomfort on or after defecation, or pain after or during sexual activity. A physical examination may reveal multiple muscle and pelvic trigger points located in the PM, OIM, LAM, and iliopsoas muscles, suggestive of myofascial pain [18].(d)Physicians are often confused by patients with a history of multiple medical conditions, including depression, anxiety, or post-traumatic stress disorder, as well as patients with common comorbidities, such as fibromyalgia, migraine, tension headache, or chronic fatigue syndrome [29].

In this work, the demographic characteristics of each group differed. Patients with vulvar and perineal pain were most unlike the other groups. This may be due to three factors: (a) a shorter median duration of pain given that the inclusion criterion was pain that had been ongoing for three months, which was defined in order to include patients with vulvodynia in this study; (b) the correlation between youth, pain during sexual intercourse, and nulliparity (Table 1); and (c) lower values for the central sensitization PP score, as a consequence of a shorter duration of pain, a lower rate of comorbidities, and/or a lower frequency of anorectal pain. These data suggest that cross-sensitization between pelvic organs may be more limited in these patients (Table 1).

Another interesting aspect was the high rate of comorbidities (36%). Comorbidities suggest a dysfunctional central mechanism [17,30,31]. Over one third of our patients showed the concomitant presence of clinical hypersensitivity syndromes, the most frequent being endometriosis.

After interviewing and examining the study patients, signs of central sensitization were noted in approximately 80% (tier 2). In these patients, the duration of pain (three years) was twice as long as in those who did not show these signs (*p* = 0.031). In CPPS patients, perineal and/or vulvar pain may be triggered by the use of tampons or by wearing tight clothing or underwear. This study suggests that vestibular allodynia/hyperalgesia is present on the physical examination of patients with CPPS due to a non-gynecologic condition in a similar percentage as in gynecologic patients (total 71%, range: 60–77%) (Table 2). Likewise, this study confirms the presence of neuropathic pain in the vast majority of patients with CPPS, regardless of its origin. Neuropathic pain was found upon examination in 84% of patients with pain on compression of the pudendal nerve and 37% of patients with positive Tinel’s sign in the clitoris (Table 2). Furthermore, nearly the entire cohort (regardless of pain origin) had trigger points upon examination of the pelvic musculature (LAM: 89%, OIM: 92%, PM: 52%) and in the pelvic girdle structures (retropubic: 86%, sacrospinous ligament: 65%, and coccyx: 67%) (Table 2). Definitively, most of these findings in women with CPPS can be explained by the pathophysiology of CPPS.

Providing physical examination information and commonalities in women exhibiting chronic pain with central sensitization and suffering from different diseases is one of the major strengths to consider in this study, thus identifying women as a group of patients with a similar prognosis and with multimodal disease management. In contrast, the present study has several limitations, resulting from its retrospective nature and the small number of patients included in the anorectal pain group. A control group without CPPS might also improve the study outcomes, since the pain mapping method is not justified in patients without pain, where the finding of trigger points is not foreseeable. Finally, the study was conducted in a gynecology department, so its results imply a lack of generalizability to men. Future research should take into account the limitations described above.

## 5. Conclusions

CPPS physical examinations can be remarkably similar, despite the underlying disease or condition causing the symptoms. Central sensitization and cross-organ sensitization are present in the vast majority of patients who do not respond to conventional therapy. Sign and symptoms such as vulvar or perineal allodynia and/or hyperalgesia, trigger points in the pelvic floor musculature, tender points in the structures related to the pelvic girdle, and neuropathic pain are frequent findings resulting from the pathophysiological process described in patients with CPPS. As a consequence, physical examination in patients with central sensitization identifies the co-existence of somatic, visceral, and neuropathic pain symptoms, thus allowing the recognition of the pelvic pain patient phenotype, which will be crucial for decision making regarding the management and prognostic factors of the disease. 

## Figures and Tables

**Table 1 jcm-11-07490-t001:** Demographic characteristics of patients with endometrioses unresponsive to conventional therapy.

	Overall N = 107	Pelvic Pain. N = 32	Anorectal Pain. N = 15	Vulvar/Perineal Pain. N = 60	*p*
**Age, years, median (range)**	41 (16–72)	42 (26–55)	47 (22–72)	38 (16–72)	**0.022**
**Pain intensity, median VAS (range)**	7 (1–10)	7.5 (5–10)	8 (1–10)	7.5 (2–9.5)	0.343
**Duration of pain, years, median (range)**	1.5 (0.3–26)	3.5 (0.3–26)	3.5 (0.5–9)	2 (0.3–24)	0.277
**Comorbidities, N (%)**	38 (36%)	17 (53%)	5 (33%)	16 (27%)	**0.040**
● Migraine	6 (6%)	3 (9%)	0 (0%)	3 (5%)	
● Fibromyalgia	10 (8%)	6 (19%)	1 (7%)	3 (5%)	
● Irritable bowel syndrome	10 (8%)	9 (28%)	0 (0%)	1 (2%)	
● Interstitial cystitis	6 (6%)	6 (19%)	0 (0%)	0 (0%)	
● Endometriosis	30 (28%)	21 (66%)	1 (7%)	8 (13%)	
**Convergences PP Score Central Sensitization, ≥5/10, N (%)**	84 (79%)	28 (88%)	14 (93%)	42 (70%)	**0.048**
**Parity**					**0.005**
● Nulliparous	52 (49%)	12 (37%)	3 (20%)	37 (62%)	
● Primi/Multiparous	55 (51%)	20 (63%)	12 (80%)	23 (38%)	
**Cesarean**	21 (20%)	9 (28%)	4 (27%)	8 (13%)	0.179
**Surgery**	73 (68%)	25 (78%)	11 (73%)	37 (62%)	0.244
**Pain location**					
● Pain during sexual intercourse	89 (83%)	22 (69%)	11 (73%)	56 (93%)	**0.006**
● Proctalgia	42 (39%)	20 (63%)	15 (100%)	7 (12%)	**<0.001**
● Urological pain	51 (48%)	19 (59%)	6 (40%)	26 (43%)	0.278

Abbreviation: N: number of patients, %: percentage; VAS: visual analog scale.

**Table 2 jcm-11-07490-t002:** Descriptive analyses of findings of patients with endometrioses unresponsive to conventional therapy.

	Overall N = 107	Pelvic Pain. N = 32	Anorectal Pain. N = 15	Vulvar/Perineal Pain. N = 60	*p*
Sensory deficit at S2–S4	71 (66%)	23 (72%)	10 (67%)	38 (63%)	0.718
Q tip test at vestibule	76 (71%)	21 66%)	9 (60%)	46 (77%)	0.322
Negative reflexes	22 (21%)	4 (13%)	5 (33%)	13 (22%)	0.245
Pain at peripheral nerves	95 (89%)	30 (94%)	15 (100%)	50 (83%)	0.107
● Pain at third segment of the pudendal nerve	90 (84%)	28 (88%)	15 (100%)	47 (78%)	0.100
● Pain at dorsal clitoris nerve	40 (37%)	12 (38%)	5 (33%)	23 (38%)	0.938
Pain in pelvic muscles	105 (98%)	32 (100%)	14 (93%)	59 (98%)	0.286
Levator ani	95 (89%)	30 (94%)	9 (60%)	56 (93%)	**0.001**
Obturator internus	98 (92%)	32 (100%)	11 (73%)	55 (92%)	**0.009**
● Medium segment	50 (47%)	16 (50%)	4 (27%)	30 (50%)	0.244
● Pelvic segment	58 (54%)	22 (69%)	7 (47%)	29 (48%)	0.142
● Ischium segment	59 (55%)	18 (56%)	6 (40%)	35 (58%)	0.438
Piriformis	56 (52%)	24 (75%)	5 (33%)	27 (45%)	**0.007**
Pain at pelvic girdle	103 (96%)	32 (100%)	15 (100%)	56 (93%)	0.196
● Retropubic	92 (86%)	28 (88%)	11 (73%)	53 (91%)	0.312
● Ischiopubic ramus	63 (59%)	21 (66%)	5 (33%)	37 (62%)	0.089
● Ischium	44 (41%)	16 (50%)	3 (20%)	25 (42%)	0.149
● Sacrospinous ligament	70 (65%)	24 (75%)	11 (73%)	35 (58%)	0.218
● Sacrum	17 (16%)	7 (22%)	2 (13%)	8 (13%)	0.542
● Coccyx	72 (67%)	25 (78%)	10 (67%)	37 (62%)	0.277

Abbreviation: N: number of patients, %: percentage.

## Data Availability

Data presented in this manuscript are available from the corresponding authors on reasonable request.

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
