# Peer review of "Understanding the Female Physical Examination in Patients with Chronic Pelvic and Perineal Pain"

_jcm, 2022, doi:10.3390/jcm11247490_

Round 1

Reviewer 1 Report

Thank you for your invitation to review this manuscript: "Understanding the physical examination of patients with 2 chronic pelvic and perineal pain". This study is important to the Gynecological filed, and provides new evidence regarding physical examination and central sensitization among CPP patients.

Few comments:

Introduction- The definition- I recommend being accurate in the definitions.

I suggest to read the following articles which may help ti the understanding of different definitions as well as the diagnosis and the physical intervention.

Grinberg, K.; Sela, Y.; Nissanholtz-Gannot, R. New Insights about Chronic Pelvic Pain Syndrome (CPPS). Int. J. Environ. Res. Public Health 202017, 3005. https://doi.org/10.3390/ijerph17093005

Grinberg, K., Weissman-Fogel, I., Lowenstein, L., Abramov, L., & Granot, M. (2019). How Does Myofascial Physical Therapy Attenuate Pain in Chronic Pelvic Pain Syndrome?. Pain research & management, 2019, 6091257. https://doi.org/10.1155/2019/6091257

The introduction is simplistic. Is necessary to add references for your statement.

There is still a lack of the rationale for conducting the study in the introduction section, and it is scarce.

Methods:

Please add more relevant details regarding the inclusion and exclusion criteria for participation in the study.

Was only the sociodemographic data presented checked? What about gender differences? For example, it is unclear whether the study included womens/ mens/ or others, it should be clarified.

Please clarify on a symbol what was the division of the groups? What about those who suffered both, where were they included?

Regarding the examination - more should be said about it, it is missing and it is worth clarifying to the reader what exactly was done and at what times.

Discussion

What is the innovation in this article? How is it different or adds to other studies?

Conclusion-

Conclusions should be written that are relevant to the current study

References

Some of the references are very old. Please update the list with the last relevant publications in this filed. For example, references number:

2,7,13,17,18

Author Response

Thank you for your invitation to review this manuscript: "Understanding the physical examination of patients with 2 chronic pelvic and perineal pain". This study is important to the Gynecological filed, and provides new evidence regarding physical examination and central sensitization among CPP patients. 

Few comments:

Introduction- The definition- I recommend being accurate in the definitions.

I suggest to read the following articles which may help ti the understanding of different definitions as well as the diagnosis and the physical intervention.

Grinberg, K.; Sela, Y.; Nissanholtz-Gannot, R. New Insights about Chronic Pelvic Pain Syndrome (CPPS). Int. J. Environ. Res. Public Health 2020, 17, 3005. https://doi.org/10.3390/ijerph17093005

Grinberg, K., Weissman-Fogel, I., Lowenstein, L., Abramov, L., & Granot, M. (2019). How Does Myofascial Physical Therapy Attenuate Pain in Chronic Pelvic Pain Syndrome?. Pain research & management, 2019, 6091257. https://doi.org/10.1155/2019/6091257

The introduction is simplistic. Is necessary to add references for your statement.

There is still a lack of the rationale for conducting the study in the introduction section, and it is scarce.

Response: The introduction has been rewritten and referenced. Following the reviewer's recommendations, a new definition has been taken from the suggested references.

Methods:

Please add more relevant details regarding the inclusion and exclusion criteria for participation in the study.

Was only the sociodemographic data presented checked? What about gender differences? For example, it is unclear whether the study included womens/ mens/ or others, it should be clarified.

Please clarify on a symbol what was the division of the groups? What about those who suffered both, where were they included?

Regarding the examination - more should be said about it, it is missing and it is worth clarifying to the reader what exactly was done and at what times.

Response: Data on inclusion and exclusion criteria have been expanded, including the exclusive participation of women in the study, references have been added to describe the pain locations (Rome IV criteria and IVSS vulvodynia criteria), and we have also added where participants with more than one pain location were assigned. Finally, the examination of each patient has been defined in more detail.

Discussion

What is the innovation in this article? How is it different or adds to other studies?

Response: The discussion has been rewriter in order to improve readability, focusing specially on the paragraphs appointed by the reviewer.

Conclusion-

Conclusions should be written that are relevant to the current study

Response: The conclusion has been rewritten.  

References

Some of the references are very old. Please update the list with the last relevant publications in this filed. For example, references number: 

2,7,13,17,18

Response: The oldest references have been removed and further references have been added to the revised text.

Reviewer 2 Report

The authors describe their cohort of patients with 

1.  The authors need to describe the significance of the chosen phenotypes of pelvic pain, rectal/anal pain, an vulvar/perineal pain.  How do their prognoses and treatments differ, and what was their rationale for choosing this stratification.  Same for central sensitization.  They should do so in the introduction.

2. Were all patients in this study female?  If so, the title should changed to understanding the physical examination in female patients with chronic pelvic and perineal pain.  This should also be noted in the abstract: Methods: "We reviewed clinical data from 107 female patient...", and inclusion criteria.  If there were males in the study, the authors need to include the number and stratify by male vs female for analysis.

3. P values <0.001 should be listed as such and not as 0.000, etc.

4. The discussion could be improved.  Early on in the discussion, the authors should discuss the implications of their findings for patient diagnosis/prognosis and treatment.  The final sentence of the current manuscript discussion "...The diagnosis of central sensitization predicts a poor, transient response to conventional treatment and can be considered a risk factor for postoperative pain" is an important point.  This should be expanded upon and this information should be provided earlier on in the discussion.  Does central sensitization guide therapy in any way?  How are these clinical examination findings clinically useful I terms of prognosis/counseling and in treatment choice?

5. The current study has several limitations, which need to be discussed in a limitations section.  These include the lack of a control group without CPPS.   What proportion of the general population has such physical examination findings?  The lack of the control group here limits the validity of the findings described. The authors should discuss the literature that The following should be clarified in the methods and addressed in limitations. Who performed the physical examinations?  Was it an individual trained in neurology?  Were multiple clinicians performing the evaluations, and would heterogeneity in physical examination create bias?  If only females were included, a limitation is the lack of generalizability to males.

Author Response

Comments and Suggestions for Authors

The authors describe their cohort of patients with 

1. The authors need to describe the significance of the chosen phenotypes of pelvic pain, rectal/anal pain, an vulvar/perineal pain.  How do their prognoses and treatments differ, and what was their rationale for choosing this stratification.  Same for central sensitization.  They should do so in the introduction.

Response: The introduction has been rewritten following the reviewer recommendations.

2. Were all patients in this study female?  If so, the title should changed to understanding the physical examination in female patients with chronic pelvic and perineal pain.  This should also be noted in the abstract: Methods: "We reviewed clinical data from 107 female patient...", and inclusion criteria.  If there were males in the study, the authors need to include the number and stratify by male vs female for analysis.

Response: All patients in this study were women, following the reviewer recommendations the title has been changed and revised in the text.

3. P values <0.001 should be listed as such and not as 0.000, etc.

Response: Following the reviewer indications, all p-value has been corrected in the revised text

4. The discussion could be improved.  Early on in the discussion, the authors should discuss the implications of their findings for patient diagnosis/prognosis and treatment.  The final sentence of the current manuscript discussion "...The diagnosis of central sensitization predicts a poor, transient response to conventional treatment and can be considered a risk factor for postoperative pain" is an important point.  This should be expanded upon and this information should be provided earlier on in the discussion.  Does central sensitization guide therapy in any way?  How are these clinical examination findings clinically useful I terms of prognosis/counseling and in treatment choice?

Response: The discussion has been rewriter in order to improve readability, focusing specially on the paragraphs appointed by the reviewer.

5. The current study has several limitations, which need to be discussed in a limitations section.  These include the lack of a control group without CPPS.   What proportion of the general population has such physical examination findings?  The lack of the control group here limits the validity of the findings described. The authors should discuss the literature that The following should be clarified in the methods and addressed in limitations. Who performed the physical examinations?  Was it an individual trained in neurology?  Were multiple clinicians performing the evaluations, and would heterogeneity in physical examination create bias?  If only females were included, a limitation is the lack of generalizability to males.

Response: A sentence with the limitations of the study has been included at the end of the discussion section, among them, the lack of a control group without CPPS and the absence of males. In the methods section it has been included that the examinations have been performed only by a clinician expert in chronic pelvic pain (experience of more than 20 years and trained at the Hospital CHU Nantes, France by Dr. Jean-Jacques Labat, urologist and neurologist).
